# Multilingual translation for zero-shot biomedical classification using BioTranslator

Hanwen Xu[1], Addie Woicik[1], Hoifung Poon[2], Russ B. Altman [3,4,5] & Sheng Wang [1] ✉

Existing annotation paradigms rely on controlled vocabularies, where each data instance is classified into one term from a predefined set of controlled vocabularies. This paradigm restricts the analysis to concepts that are known and well-characterized. Here, we present the novel multilingual translation method BioTranslator to address this problem. BioTranslator takes a user-written textual description of a new concept and then translates this description to a non-text biological data instance. The key idea of BioTranslator is to develop a multilingual translation framework, where multiple modalities of biological data are all translated to text. We demonstrate how BioTranslator enables the identification of novel cell types using only a textual description and how BioTranslator can be further generalized to protein function prediction and drug target identification. Our tool frees scientists from limiting their analyses within predefined controlled vocabularies, enabling them to interact with biological data using free text.

High-throughput techniques have created ever-expanding repositories of omics datasets that continually uncover new features of biological systems[1–7]. One of the first analysis steps for a new dataset is annotation[8–12], where each data instance (e.g., a cell) is manually or automatically classified into one term (e.g., cell type) from a predefined set of controlled vocabularies (CVs)[13–19]. CVs serve as the anchors to integrate datasets generated by different labs[20,21], and later enable biologists to immediately locate the pertinent data instances according to the corresponding CV[22]. However, this CV-based annotation paradigm does not extend well to the analysis of new findings that do not fall into the categories of any existing CVs. Creating new CVs is time-consuming and requires substantial domain knowledge to summarize novel terms using a short phrase that does not semantically overlap with existing CVs[19,23–25].

To address this problem, we propose BioTranslator, a novel multilingual translation method that takes a user-written textual description (e.g., a sentence) as input and then identifies relevant non-text data instances according to this description (Fig. 1a). As a result, BioTranslator enables scientists to find data instances related to any topic they are interested in without being restricted by existing CVs. An

analog of the difference between CV-based analysis and BioTranslator is the old-fashioned Yahoo directory[26] and the transformative Google search engine:[27] the Yahoo directory only supports the retrieval of websites using pre-defined hierarchical categories, whereas Google enables text-based search to find any websites that match the free text.

Notably, our method does not simply use text-based similarity to find the semantically similar CV for a given textual description[28–30] and then yield the data instances from that CV. In contrast, we translate user-written text to a non-text biological data instance, such as a gene expression vector, and then find other similar data instances in the biological data space. This process is similar to cross-modal learning in other domains[31–35], such as image caption generation[36]. These existing cross-modal learning approaches are bilingual, where only two modalities are translated to each other (e.g., image and text in caption generation[36], protein sequence and text in ProTranslator[37]). In contrast, BioTranslator is a multilingual translation framework, where different modalities of biomedical data are all mapped to a shared latent space, which further enables new analyses and results that cannot be achieved using a bilingual framework. The key technique supporting our method is to fine-tune large-scale pretrained language

[1]School of Computer Science and Engineering, University of Washington, Seattle, WA, USA. [2]Microsoft Research, Redmond, WA, USA. [3]Department of Bioengineering, Stanford University, Stanford, CA, USA. [4]Department of Genetics, Stanford University, Stanford, CA, USA. [5]Chan Zuckerberg Biohub, San Francisco, CA, USA. ✉e-mail: swang@cs.washington.edu

**a**

**b** **c** **d** **e**

models using existing biomedical ontologies based on a contrastive learning loss[38]. We then project the encoded textual description to the biological data space using paired data constructed from existing CV-based annotations[13,39–42].

This multilingual translation framework enables BioTranslator to perform zero-shot classification, where we can classify test instances into never-before-seen classes that do not have any annotated training instances. We first show that BioTranslator can annotate proteins to a novel function by only using the textual description of this function, suggesting the possibility to expand the Gene Ontology (GO) and its annotations using text[13,43,44]. We then illustrate how BioTranslator can accomplish the reverse task: generating textual descriptions to describe the function of a set of genes, even if this set does not have any substantial enrichment with existing gene sets. On Tabula Muris[10],

**Fig. 1 | BioTranslator overview. a** BioTranslator takes a user-written text as input and then translates it into non-text biological data. BioTranslator uses a multilingual framework and can be applied to diverse applications. For example, in the context of cell type classification, BioTranslator embeds cell type using cell type textual description obtained from the Cell Ontology and cells using gene expression. It then co-embeds these two modalities using existing annotations. When users query a new cell type using textual description, BioTranslator will identify relevant cells in this co-embedding space. Created with BioRender.com. **b**–**e** Box plots comparing text-based similarity with annotation-based GO similarity (**b**), graph-based GO similarity (**c**), annotation-based pathway similarity (**d**), and annotation-based cell type similarity (**e**). The text-based similarity is calculated using the embeddings of textual descriptions. The annotation-based similarity is calculated using binary vectors over proteins (**b**), genes (**d**), or cells (**e**). The graph-

based similarity is calculated using the shortest distance on the GO graph. For each box plot, the minima is the lower quartile (Q1) – 1.5*interquartile range (IQR), the maxima is the upper quartile (Q1) + 1.5*interquartile range (IQR), the center is median (Q2), the bounds of box are Q1 and Q3, the whiskers are from the minima to Q1 and from Q3 to the maxima. 2000 GO terms were randomly chosen and $n = 1998074, 243, 313, 322, 45$ GO term pairs examined when the annotation-based similarity is in 0–0.2, 0.2–0.4, 0.4–0.6, 0.6–0.8, and >0.8 (**b**). 2000 GO terms were randomly chosen and $n = 535, 541, 346, 228, 1997350$ GO term pairs examined when the graph-based GO distance is 1, 2, 3, 4, and >4 (**c**). $n = 1676749, 4238, 1792, 1169, 582$ pathway pairs examined when the annotation-based similarity is in 0–0.2, 0.2–0.4, 0.4–0.6, 0.6–0.8, and >0.8 (**d**). $n = 2424110, 705, 4168, 3166, 4376$ cell type pairs examined when the annotation-based similarity is in 0–0.2, 0.2–0.4, 0.4–0.6, 0.6–0.8, and >0.8 (**e**).

---

Tabula Sapiens[11], and Tabula Microcebus[12], we demonstrate how BioTranslator can obtain an average AUROC of 0.90 for classifying cells to a specific subtype using only a textual description, with no access to annotated cells or marker genes of that subtype. BioTranslator further enables the identification of marker genes for these novel cell types by only using the user-written textual description. Finally, we demonstrate BioTranslator's ability to translate between two modalities without using any paired data between them. This is achieved through translating both modalities to a third modality, which is enabled using the multilingual translation framework in BioTranslator. We found that BioTranslator obtained promising prediction performance on drug target identification[45,46], phenotype-pathway association[47,48], and phenotype-gene association[49]. BioTranslator, which bridges text and biomedical data using large-scale pretrained language models, facilitates the analysis of new discoveries and can be broadly applied to diverse biological tasks and domains.

## Results
### Overview of BioTranslator
BioTranslator enables users to find relevant data instances from a large dataset for a new class they are interested in. BioTranslator does not need any annotated training instances for this new class. The only information BioTranslator needs is a user-written textual description for this class. This text could be a few keywords or sentences, which is designed similarly to the input to the Google search engine. We further require that a proportion of the data instances in this large dataset have been annotated to controlled vocabularies. The remaining data instances do not need to be annotated to any controlled vocabularies and can be later annotated to any new classes described by the user.

In a nutshell, BioTranslator learns a cross-modal translation to bridge text data and non-text biological data (Fig. 1a). At the training stage, BioTranslator first constructs a paired text to non-text training dataset using existing annotated data instances. This is analogous to a parallel corpus dataset between two languages in machine translation[50]. For example, in the context of cell type classification, each text datum is the cell type name or description obtained from the Cell Ontology[39]. Each non-text biological datum is a gene expression vector. Using this paired training dataset, BioTranslator projects the text data and non-text biological data into the same embedding space based on a contrastive-learning-based loss function[38]. To integrate different modalities, BioTranslator exploits PubMedBERT[51] to model the textual description and fine-tunes it using ontologies from diverse domains. At the test stage, BioTranslator first embeds a new class into this space according to the user-written textual description and then annotates nearby biological data instances to this new class.

This procedure does not need any annotated samples for this new class, thus enabling zero-shot classification. Besides, this multilingual translation framework also enables other novel applications, such as generating textual description for biological data instances and text-based biomedical discovery interpretation. Moreover, we illustrate

how BioTranslator can be broadly applied to diverse applications, including cell type classification and protein function prediction. The non-text data for these three applications could be gene expression vectors, protein sequences and drug SMILES representations, respectively. The training text data of all these three applications are obtained from existing databases[13,39–41,52].

### Class textual description similarity reflects annotation similarity
The assumption of BioTranslator is that classes that are annotated to similar instances should have similar textual descriptions. We used GO-based protein function annotation, where each GO term is a class, to verify this assumption. Specifically, we investigated the correlation between the annotation-based GO term similarity and the text-based GO term similarity. The annotation-based GO term similarity represents each GO term using the binary vector over its protein annotations (see Methods). The text-based GO term similarity represents each GO term using its textual description. These two similarities showed a substantial correlation ($p = 2.28e-207$ using the ANOVA test). This demonstrates that GO terms that have similar textual descriptions tend to have similar gene annotations (Fig. 1b). We next investigated the similarity between the text-based GO term similarity and graph-based GO similarity derived from the Gene Ontology graph[53]. We found that terms that have higher textual similarity are much closer to each other on the graph (Fig. 1c). GO graph has been successfully used to assist function annotation[53–55]. Therefore, the strong agreement between text-based GO similarity and graph-based GO similarity further suggests that terms with similar textual descriptions are likely to be annotated to the same proteins. We observed similar consistency on cell type annotation and pathway annotation (Fig. 1d, e, Supplementary Fig. 1), validating the assumption of BioTranslator in diverse applications.

### Accurate annotation of new functions using BioTranslator
We first sought to systematically evaluate BioTranslator using GO-based protein function prediction[37,53,56]. Here, we held out all the protein annotations of a GO function and asked BioTranslator to recover these proteins using only the textual description of this function. This setting simulates the process where a user searches for proteins related to a new function they thought might exist by writing a textual description. BioTranslator translates user-written textual description to a protein representation mixed from network, sequence, and description information (see Methods).

We summarized the results on three GO domains—biological process (BP), molecular function (MF), and cellular component (CC)—across five datasets in Fig. 2a–e and Supplementary Figs. 2–6. We first compared BioTranslator to three conventional text modeling approaches TF-IDF[29], Word2Vec[28], and Doc2Vec[57]. These approaches have been widely used to model biological text[58–60], but none of them are able to model contextual interdependence as pretrained language models do. BioTranslator obtained on average 0.10 (BP), 0.13 (MF),

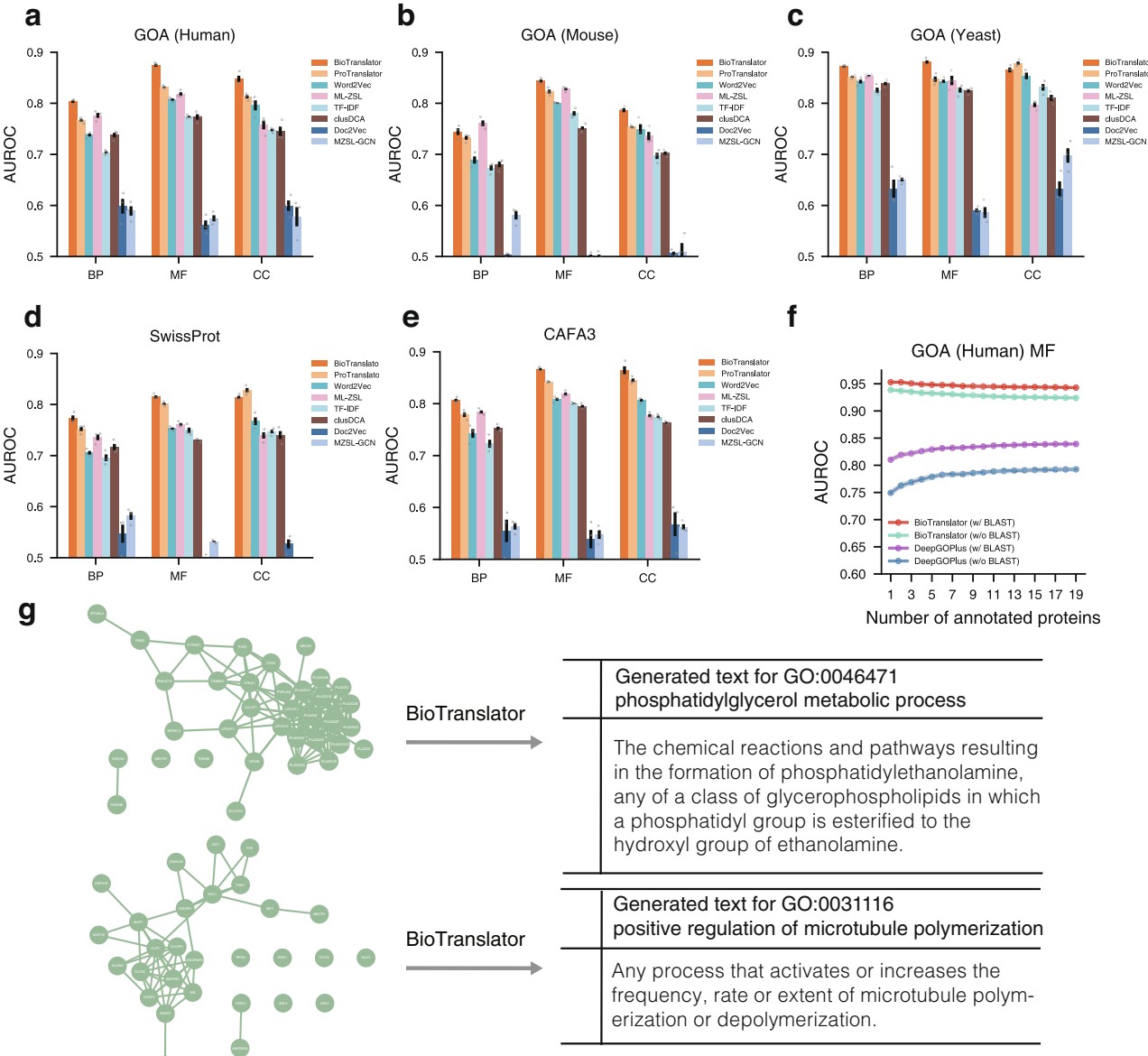

**Fig. 2 | Performance of BioTranslator on annotating novel protein functions.** **a**–**e** Bar Plots showing the AUROC of BioTranslator, ProTranslator, TF-IDF, Word2vec, Doc2vec, ML-ZSL, MZSL-GCN, and clusDCA on GOA (Human), GOA (Mouse), GOA (Yeast), SwissProt, and CAFA3. All proteins annotated by test functions were excluded. BP biological process, MF molecular function, CC cellular component. Error bar represents standard errors across 3-fold cross-validation and its center is mean value. **f** Plot comparing the AUROC of using BioTranslator (w/ BLAST), BioTranslator (w/o BLAST), DeepGOPlus (w/ BLAST), and DeepGOPlus (w/o BLAST) for function annotation on GOA (Human) molecular function domain. The protein annotation number of functions is from 1 to 20. The error bands represent the standard errors across threefold cross-validation. **g** The generated text from BioTranslator given the proteins of phosphatidylglycerol metabolic process (GO:0046471) and the proteins of positive regulation of microtubule polymerization (GO:0031116). Proteins are connected using a protein-protein interaction network from STRING for visualization. The network structure is not used to generate the text.

and 0.12 (CC) absolute improvements in terms of AUROC on the five datasets compared to these approaches, indicating the superiority of using large-scale pretrained language models. We then compared BioTranslator to ProTranslator[61], which exploits the same pretrained language models to embed textual description. ProTranslator is bilingual since it only embeds textual descriptions from the GO. In contrast, BioTranslator is developed as a multilingual framework by using textual descriptions from 225 ontologies to fine-tune Pub-MedBERT. BioTranslator substantially outperformed ProTranslator, indicating the promising performance from a multilingual framework. clusDCA[53], ML-ZSL[62], and MZSL-GCN[63] classified proteins into novel functions using the GO graph. We found that BioTranslator out-

performed these graph-based approaches, showing the advantage of using textual descriptions over using the GO graph to annotate novel functions. Moreover, graph-based approaches require the queried function to be one of the terms in the GO graph, whereas our method can be applied to any novel function using only its textual description. We further studied the few-shot setting[53,64], where each test GO term only has very few annotated proteins in the training data. Since existing protein function prediction methods can be applied to this setting, we compared BioTranslator to the state-of-art protein function prediction approach DeepGOPlus[65]. We found that BioTranslator substantially outperformed DeepGOPlus for GO terms with very few annotations in the training data (Fig. 2f, Supplementary Figs. 7, 8).

### Generating a textual description for a set of proteins

Since BioTranslator can be regarded as a machine translator that translates between text data and non-text biological data, it can also be applied to the reverse task of generating a textual description for a biological data instance. This task can expand the curation of controlled vocabularies and textual definition in existing ontologies[13,19,66]. Here, we provided a set of proteins that were associated with the same GO term to BioTranslator and asked it to generate a textual description for them. The ground truth output is the curated textual description of that GO term. We compared our generated textual description to the curated GO term description on GOA (human) and obtained a 0.32 BLEU (percentage of 1-g, 2-g, 3-g, 4-g overlaps between two text instances), which is much higher than the 0.26 BLEU from ProTranslator[61]. 0.32 BLEU is considered as desirable text generation performance according to the state-of-the-art machine translation model[67]. We found that many generated descriptions were biologically meaningful and accurate by comparing to the curated GO term description (Fig. 2g, Supplementary Fig. 9), suggesting that our method can assist GO expansion by providing descriptions for new protein sets.

Finally, we evaluated BioTranslator using different pretrained language models[51,68–71], which were trained using different corpora and model architectures. We observed consistent performance by using different pretrained language models, indicating the robustness of BioTranslator (Supplementary Fig. 10). We found that PubMedBERT[51], which performed domain-specific pretraining, achieved the best performance among all pretrained language models. This is consistent with prior findings that domain-specific pretraining is advantageous because it prioritizes in-domain text modeling[51,72]. Therefore, we use PubMedBERT as the default model in the following analysis. Moreover, we found that directly using pretrained language models without fine-tuning performed substantially worse than BioTranslator, indicating the effectiveness of exploiting biomedical ontologies to fine-tune the language model.

### Novel cell type discovery using BioTranslator

The improvement of the domain-agnostic BioTranslator over the domain-specific ProTranslator motivates us to further evaluate BioTranslator on other applications. We then sought to study how BioTranslator can be used to advance cell type classification through translating text to gene expression data. Here, we obtained the single-cell RNA-seq data from Tabula Muris[10], Tabula Sapiens[11], and Tabula Microcebus[12], and the textual description of cell types from the Cell Ontology[39]. We held out all cells of a specific cell type in the training data and asked BioTranslator to identify these cells using only the textual description of this cell type, simulating the process where users want to find cells of a novel cell type by only providing a short textual description (Fig. 3a–c, Supplementary Fig. 11). Almost none of the state-of-the-art cell type classification methods are able to classify cells to a novel cell type without knowing any of its cells or marker genes. In contrast, our method, using only a short textual description, obtained 0.90 AUROC on average when 50% of the cell types were unseen. After confirming the superior performance of BioTranslator in the cross-validation setting, we next evaluated the more applicable but also more challenging cross-dataset classification setting (Fig. 3d, e). We used BioTranslator to perform cell type classification across eight different datasets and again observed a prominent result using BioTranslator. For example, the AUROC on unseen cell types was >0.90 when trained on Tabula Sapiens and tested on Tabula Microcebus. Taken together, these results confirm that BioTranslator can annotate novel cell types with high accuracy across datasets by using only a concise textual description and no access to marker genes.

Next, we investigated whether our method can identify marker genes for the cell type that we have never seen in the training data using only its textual description. A recent method OnClass[73] can identify marker genes for all cell types in the Cell Ontology using only the Cell Ontology hierarchical graph[39]. Despite its successful applications, OnClass requires that cell type to be one of the controlled vocabularies in the Cell Ontology. In contrast, BioTranslator can accurately identify marker genes for a cell type that is not in the Cell Ontology, as long as users have provided a textual description for it, substantially expanding the scope of cell type discovery (Fig. 3f). By using the gene embedding learned in the protein function prediction as gene representation, BioTranslator can also accurately identify marker genes (Supplementary Fig. 12). We further examined the performance of using cell type names instead of textual descriptions and observed a slightly worse but still prominent performance, indicating the robustness of our method when only less informative user-written text is available (Fig. 3g, Supplementary Fig. 13).

Finally, we construct a cell type marker gene network using the textual embeddings derived from our method (Fig. 3h). In this network, each node is a cell type or a gene and each edge connects two nodes that have similar textual embeddings. Notably, this text-based comparison enables us to directly assess the similarity between nodes of different types, such as a cell type and a gene. By using the path distance on this cell type marker gene network, we are able to obtain a 0.82 AUROC in identifying marker genes. We used a text-based node set enrichment analysis approach to find the enriched GO term for each community in the network. For example, the proT cell type is in a community that is enriched with cytokine production[74], and many of its identified genes, such as *RAG1*, *CD3E*, *CD44*, and *ID2*, can be verified by literature[75–78]. This cell type marker gene network enables users to query and visualize marker genes using textual descriptions.

### BioTranslator enables prediction between modalities without paired data

One critical improvement of a multilingual translator over a bilingual translator is the ability to translate between two languages through a third language, thus circumventing the requirement of paired corpora between all language pairs. To verify the benefit of our multilingual translator, we investigated whether BioTranslator can achieve accurate predictions between two modalities without paired data. Specifically, we used BioTranslator to co-embed drugs, phenotypes, genes, and pathways into the low-dimensional space using their associated textual descriptions (see Methods). We then predicted drug-target interaction, phenotype-gene association, and phenotype-pathway association using independent sets of drugs and phenotypes that we have never seen textual descriptions for (Fig. 4a, b).

We found that BioTranslator obtained promising prediction performance on these three tasks without using any paired data between two modalities (Fig. 4c–f). Unlike BioTranslator, comparison approaches require paired data to perform the prediction tasks. We compared BioTranslator to various supervised comparison approaches that were trained on different proportions of paired data. In addition to being the only unsupervised approach, we found that BioTranslator was better than the supervised approaches on two tasks (Pathway2Phenotype and Gene2Drug (GDSC)). These two tasks have the smallest number of training pairs, suggesting BioTranslator's superior performance in the low-resource setting. Although BioTranslator was outperformed by supervised approaches that used class features on two larger collections, Gene2Drug (STITCH) and Gene2Phenotype, BioTranslator was still better than the supervised approach that does not use class features. Our experiments simulate the real-world settings where a new drug is discovered and we only know its SMILES ID, but do not have access to its textual description. Bilingual frameworks such as ProTranslator cannot be used to predict the new drug's target. In contrast, BioTranslator is able to predict its target without seeing any paired drug target data, demonstrating the superiority of a multilingual translation system. Since there is no well-curated drug-pathway association benchmark, we illustrated a top prediction made by

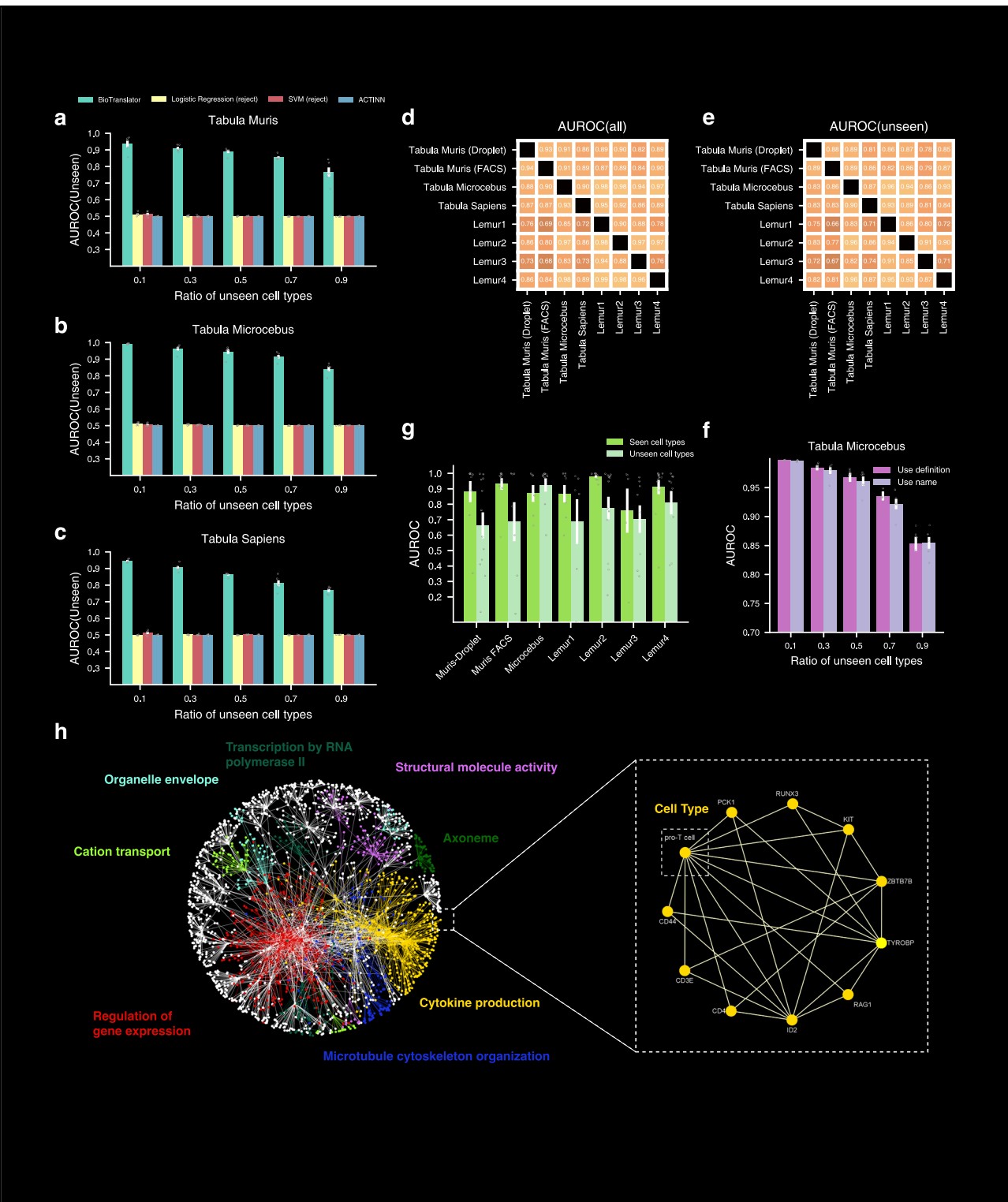

**Fig. 3 | Performance of BioTranslator on annotating novel cell types. a–c** Bar plots comparing BioTranslator to existing methods in terms of AUROC in Tabula Muris (Droplet), Tabula Microcebus, and Tabula Sapiens. *x*-axis shows the proportions of unseen cell types in the test set. Error bar shows the standard error in 5-fold cross-validation. Center of the error bar is the mean value. **d**, **e** Heatmaps showing the performance of BioTranslator in cross-dataset validation, in terms of AUROC on all cell types (**d**) and AUROC on unseen cell types (**e**). Columns are the test set and rows are the training set. **f** Bar plot showing the AUROC performance of marker gene identification on seen cell types and unseen cell types. *n* = 9, 10, 6, 7, 9, 5, 13 seen cell types examined on Tabula Muris (Droplet), Tabula Muris (FACS),

Tabula Sapiens, Tabula Microcebus, Lemur1, Lemur2, Lemur3, and Lemur4. *n* = 12, 7, 9, 6, 13, 8, 9 unseen cell types examined on Tabula Muris (Droplet), Tabula Muris (FACS), Tabula Sapiens, Tabula Microcebus, Lemur1, Lemur2, Lemur3, and Lemur4. Error bar represents the standard error across the examined cell types and the center of the error bar is the mean value. **g** Bar plot comparing the performance of BioTranslator using cell type textual descriptions to the performance using cell type names. Error bar is the standard error in fivefold cross-validation. Center of the error bar is the mean value. **h** Cell type marker gene network constructed using the textual embeddings from BioTranslator. Eight communities are detected and each is associated with a GO term using text-based GO enrichment.

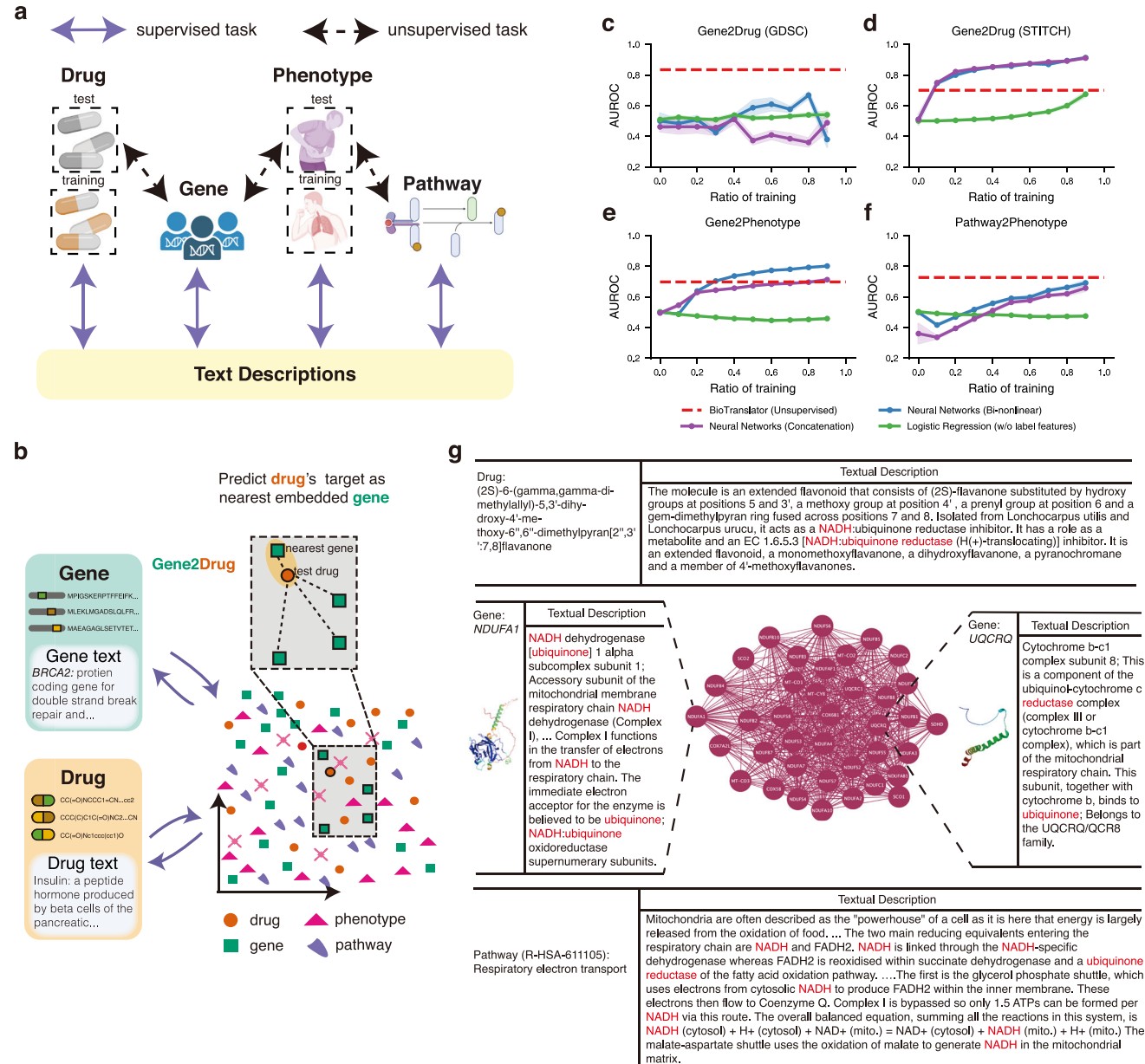

**Fig. 4 | Performance of BioTranslator on prediction without paired data.**
**a** Diagram of BioTranslator's multilingual translation framework. During the training stage, drugs, genes, phenotypes and pathways were translated into the text embedding space. BioTranslator then enables drug-target interaction prediction, phenotype-gene association prediction, and phenotype-pathway association prediction using this co-embedding space without access to the paired data between two non-text modalities. Created with BioRender.com. **b** Flowchart showing how BioTranslator predicts association between different modalities. The target of a given drug is predicted as its nearest gene in the embedding space. BioTranslator does not need to use the textual description of test drugs and phenotypes, but only needs their biological features (e.g., SMILES for drugs, human phenotype ontology for phenotypes). **c**–**f** cross-modal translation between drug and gene on GDSC (**c**), drug and gene on STITCH (**d**), gene and phenotype (**e**), and pathway and phenotype

(**f**). BioTranslator is an unsupervised approach that does not use any paired data between two modalities. In contrast, neural networks and logistic regression models are trained using paired data between two modalities and the ratio of the training data is shown on the x-axis. Neural networks combine both sample features and class features through bi-nonlinear model or concatenation. Logistic regression does not use class features. The textual descriptions of all test drugs, genes, phenotypes, and pathways are never used during the training or the test stage of the comparison approaches. Biological features of test drugs and phenotypes were also not seen during BioTranslator's training. The error bands represent the standard errors across 5-fold cross-validation. **g** The overlap words between the curated textual description of the drug, the predicted pathway, and two of the genes in that pathway. The drug description has not been used to train the BioTranslator model.

BioTranslator, where BioTranslator annotated the pathway 'Respiratory electron transport' to a selected drug (Fig. 4g). We found that the descriptions of drugs, genes, and pathways show similar words, interpreting the prediction made by BioTranslator. These biologically informative words verify the promising performance of BioTranslator, deepening our understanding of the biological system.

## Discussion

We have presented BioTranslator, a machine learning framework to annotate new biological discoveries. BioTranslator can find relevant data instances using only a short textual description. The key idea behind BioTranslator is to fine-tune large-scale pretrained language models using biomedical ontologies from diverse domains. We have

demonstrated its promising prediction performance on protein function prediction, cell type classification, drug target identification, and phenotype pathway prediction. The prediction ability of BioTranslator can facilitate future data curation and the expansion of existing controlled vocabularies.

Based on the current limitations of BioTranslator, there are a few future directions we would like to pursue. First, BioTranslator is currently evaluated and trained using expert-written textual descriptions from expert-written ontology descriptions. Since users of our tool might not have the expertise to provide a high-quality description, we will adopt text normalization methods[79] to make BioTranslator compatible with noisy text, plain language text, and short text. Second, since the performance of BioTranslator crucially relies on annotations of existing CVs, it is beneficial to refine and expand existing CVs using BioTranslator. We will identify key phrases from the high contribution words from BioTranslator and generate new CVs using these phrases through collaboration with human experts.

BioTranslator substantially advances the previous work ProTranslator by proposing a novel multilingual translation framework. In particular, ProTranslator is a bilingual translation framework, which requires access to paired data between two modalities (e.g., protein sequence and protein function text). In contrast, BioTranslator is a multilingual translation framework that jointly embeds text and multiple biomedical modalities into the same space. In traditional machine translation research[80], multilingual translation is a substantial improvement over bilingual translation due to three reasons. First, it does not require access to paired data between any two languages. Second, it effectively utilizes data from all languages to translate between any two languages. Third, it can better model low-resource languages (e.g., endangered languages) that have limited paired data with other languages. Likewise, our multilingual translation framework BioTranslator also enables analyses and results that cannot be obtained using ProTranslator for the same three reasons. First, we showed that BioTranslator can predict drug-target interaction, phenotype-pathway association, and phenotype-gene association without access to the corresponding paired data. Second, BioTranslator outperformed ProTranslator on protein function prediction, indicating the effectiveness of integrating ontologies from diverse domains. Third, we observed that BioTranslator's performance improvement relative to comparison approaches is larger on the smaller collection of phenotype-pathway associations than on the larger collection of phenotype-gene associations. Finally, compared to literature mining approaches that focus on existing phrases in scientific paper text[81–83], the sentence generated by our method can be a novel sentence, and thus can better describe new discoveries that have never been covered in existing literature or controlled vocabularies.

## Methods

### BioTranslator Model

BioTranslator is a multilingual translation framework between text data and biological data. The input of BioTranslator is a textual description and the output is a biological data instance. To train BioTranslator, we construct pairs of textual descriptions and biological data instances using existing biomedical ontologies. The text is the description of a class node in a biomedical ontology and the biological data instance is the data point that has been curated to this class. For example, the description of a GO term from the Gene Ontology can be used as the textual description and the annotated proteins to that GO term are the biological data points. BioTranslator will then learn a mapping from GO textual description to a protein sequence. One textual description could be paired to multiple biological data instances (e.g., different cells could be classified to one cell type), and one biological data instance can be paired to multiple textual descriptions (e.g., one protein could be classified into different Gene Ontology terms).

The first step in BioTranslator is to jointly embed textual descriptions from diverse biological domains. Our previous study ProTranslator has shown that large-scale pretrained language models can offer high-quality sentence embeddings in the protein function prediction. We further enhanced the quality of textual description embeddings by fine-tuning PubMedBERT on 225 existing biomedical ontologies[19,66,84], which have 2,010,648 textual descriptions. The mathematical formulations could also pinpoint the necessity of integrating information from multiple ontologies. Given one instance with feature $\mathbf{F}$ and the unseen class with textual description embedding $\mathbf{Y}_{\text{unseen}}$, our classifier should be able to approximate $P(\mathbf{Y}_{\text{unseen}}|\mathbf{F})$. We can expand it as:

$$P(\mathbf{Y}_{\text{unseen}}|\mathbf{F}) = P(\mathbf{Y}_{\text{unseen}}|\mathbf{Y}_{\text{seen}})P(\mathbf{Y}_{\text{seen}}|\mathbf{F}), \tag{1}$$

where fitting $P(\mathbf{Y}_{\text{seen}}|\mathbf{F})$ denotes the learning process of our bi-nonlinear model. Then fine-tuning the text encoder on multiple ontologies corresponds to estimate $P(\mathbf{Y}_{\text{unseen}}|\mathbf{Y}_{\text{seen}})$, where classes with similar textual descriptions tend to be co-located in the embedding space. In our implementation, we used PubMedBERT with [CLS] readout function to initialize the text encoder in BioTranslator. We fine-tuned BioTranslator text encoder by contrastive learning[38]. Textual descriptions of ontology neighbor nodes were used as positive samples and other pairs were used as negative samples. We calculated the cosine similarity of each pair and used cross-entropy as the loss function. In the training process, we used Adam as the optimizer and set the learning rate to $1 \times 10^{-5}$. To reduce the memory cost, we set the batch size to 16 and the maximum length of textual descriptions to 256 words.

The second step is to learn a mapping between textual descriptions to non-text biological instances. BioTranslator utilized deep neural networks to handle different types of biological instances, including gene expression vectors, protein networks, and protein sequences. Let one biological instance be represented by $k$ different features. We first embedded those $k$ different features with $k$ separate fully connected neural networks, then we produced $k$ embeddings listed as $\mathbf{F}_{i,1}, \mathbf{F}_{i,2}, ..., \mathbf{F}_{i,k}$, where $\mathbf{F}_{i,j} \in \mathbb{R}^{h_j}$ and $i$ was the index of samples. The hidden dimension of each embedding was $h_j$. We then concatenated these feature embeddings together into a combined feature vector $\mathbf{F}_i \in \mathbb{R}^{\sum_{j=1}^{k} h_j}$.

We trained BioTranslator using pairs of text and non-text data extracted from existing controlled vocabularies. BioTranslator took the textual description embeddings and biological instance features as input and annotated the biological instances to terms in controlled vocabularies. To formulate the training procedure, we used $\mathbf{A}_i \in \mathbb{R}^c$ to represent the annotation of sample $i$, where $\mathbf{A}_i$ is a binary vector and $c$ represents the size of controlled vocabularies. $\mathbf{A}_{i,j} = 1$ only when sample $i$ can be annotated to term $j$. We calculated the cross-entropy loss and optimize the parameters in BioTranslator using the Adam optimizer as:

$$Loss_{BioTranslator} = \sum_{i=1}^{n} \sum_{j=1}^{c} \left( -\mathbf{A}_{i,j} \log \frac{1}{1 + e^{-(\mathbf{F}_i)^T \mathbf{W} \mathbf{Y}_j}} - (1 - \mathbf{A}_{i,j}) \log \frac{e^{-(\mathbf{F}_i)^T \mathbf{W} \mathbf{Y}_j}}{1 + e^{-(\mathbf{F}_i)^T \mathbf{W} \mathbf{Y}_j}} \right), \tag{2}$$

where $\mathbf{W}$ could be optimized in the training and $\mathbf{Y}_j \in \mathbb{R}^{d_{bio}}$ was the textual description embeddings produced by our text encoder using a contrastive learning loss. $d_{bio}$ was fixed to 768.

After training the model, BioTranslator was able to annotate any novel class given a textual description. It first used the text encoder to embed the text data of the novel class into $\mathbf{Y}_{\text{novel}}$ and then embedded the input features into $\mathbf{F}_q$. The probability $p_{\text{novel}}$ was calculated using

the following formulation:

$$p_{novel} = \frac{1}{1 + e^{-(\mathbf{F}_q)^T \mathbf{W} \mathbf{Y}_{novel}}}. \tag{3}$$

Furthermore, the above multilingual translation framework supported translations between different non-text modalities. For example, the probability of one gene being paired with one drug couldbe calculated as:

$$p_{novel} = \frac{1}{1 + e^{-(\mathbf{F}_{gene})^T \mathbf{W}_{gene} \mathbf{W}_{drug}^T \mathbf{F}_{drug}}}, \tag{4}$$

where $\mathbf{F}_{gene}$ and $\mathbf{F}_{drug}$ denoted the input features of that gene and drug. $\mathbf{W}_{gene}$ was learned in the translations from genes to text and $\mathbf{W}_{drug}$ was learned by translating drugs to text.

### Text-based, graph-based, and annotation-based similarity

The text-based similarity between cell types, GO terms or pathways was obtained by calculating bilingual evaluation understudy (BLEU) score[85] of the textual descriptions. BLEU score is the percentage of 1-g, 2-g, 3-g, and 4-g overlaps between two text samples, ranging from 0–1. BLEU is 1 when two samples are identical. The graph-based distance was calculated by finding the shortest path between cell types or GO terms which were connected according to the "is_a" and "part_of". We obtained the annotation vector of a biological term and then used the cosine similarity to calculate the annotation-based similarity between them. The binary annotation vector was defined by setting its component to 1 when this component corresponded to this biological term or the ancestor of this term in the terminology hierarchy and 0 otherwise. For the annotation-based pathway similarity, we calculated the Jaccard similarity between the gene sets of two pathways.

### Protein function prediction

In protein function prediction, we collected the textual descriptions of each GO term by concatenating the text in "name" and "definition" fields in the GO. We evaluated our method on GOA (Human)[13], GOA (Mouse)[13], GOA (Yeast)[13], SwissProt[44], and CAFA3[43] datasets. These datasets contain the protein sequence features and its annotations. We further collected the protein description features from GeneCards[86,87] and the protein network features from the STRING database v9.1 and v10[88]. We used the protein network embeddings produced by Mashup[49] as the protein network features. We adopted the 3-fold cross-validation to evaluate BioTranslator, where leaf nodes were selected as the unseen functions and any training samples annotated by these nodes were excluded to avoid information leakage. We adopted the area under the receiver of the characteristic curve (AUROC)[89] as the metric to evaluate the performance. We used scikit-learn Python package (v0.24.2) to calculate the AUROC values. We constructed seven baselines for comparison. ProTranslator directly used PubMedBERT without using biomedical ontologies to fine-tune the model. We also embedded the textual descriptions with Term Frequency–Inverse Document Frequency (TF-IDF)[59], Word2Vec[28], and Doc2Vec[57]. We further compared BioTranslator to the graph-based approach clusDCA[53] by replacing the text vectors with the ontology network vectors produced by clusDCA. We also compared our method with two multi-label zero-shot learning approaches, ML-ZSL[62] and MZSL-GCN[63]. ML-ZSL learns a propagation matrix in the label semantic space. MZSL-GCN further uses label co-occurrence to model the label correlation matrix. The learned graph propagation mechanisms can be used to predict unseen labels. These two methods were developed on image classification tasks. To apply them to our setting, we exploited their architectures to propagate information between labels and combined them with our biological instance encoder.

BioTranslator embedded each GO term using its textual description. BioTranslator embedded each protein using the protein sequence, the protein textual description and the protein network features. For the sequence feature with length $L$, we adopted extra neural network layers from DeepGOPlus to embed the sequence features. We used the one-hot encodings to embed sequences. Then we obtained the sequence encodings using 1-d convolution with 16 different kernel sizes: 8, 16, 24, …, 128. The number of filters for each size was set to 512. We followed DeepGOPlus to add the max pooling layer for each convolution layer separately. The kernel size of the max pooling layer was set to the same as the tensor length after each 1-d convolution operation, resulting in 16 tensors with the size of 512. These tensors were concatenated into one tensor with the size of 8192 as the sequence feature embeddings. The hidden sizes of the embedded vectors for sequence, description and network features were all set to 1500. For description features, we used PubMedBERT to calculate the feature embeddings. We trained our model and all the baselines for 30 epochs.

In the task of generating text for a given protein set, BioTranslator utilized Textomics[90], a recent text generation method, to integrate the $k$ nearest existing GO terms based on Jaccard similarity between gene sets. In our implementation, we set $k$ to 5. Then we utilized the pre-trained T5[91] model for fine-tuning. In the text generation stage, we evaluated our method using the standard machine translation method BLEU[85] score by comparing the generated text and the ground truth text from the GO. We used NLTK Python package (v3.7) to calculate the BLEU scores in generating textual description. The gene set for each function was extracted from the GOA (Human) dataset. We split the dataset according to the GO functions. We randomly chose 70% of the GO functions to construct the training gene sets and the remaining 30% of functions to construct the validation gene sets. The learning rate, batch size, and training epochs were set to 0.0001, 16 and 15, respectively. The number of warmup steps in training was 200. When generating text, the number of beams, the repetition penalty and the length penalty were 2, 2.5 and 1.0, respectively.

### Cell type discovery

In single cell analysis, our paired text data and biological data instance were the textual description of a Cell Ontology term and the gene expression vector of a corresponding cell. The Cell Ontology was downloaded from The OBO foundry[19]. We constructed a Cell Ontology graph using the 'is_a' relationship. We collected textual descriptions using the "name" and "definition" fields from 2330 Cell Ontology terms. Tabula Sapiens was downloaded from https://tabula-sapiens-portal.ds.czbiohub.org/whereisthedata[11]. Tabula Microcebus was download from https://figshare.com/projects/Tabula_Microcebus/112227[12]. Tabula Muris (FACS), Tabula Muris (Droplet) and Lemur 1-4 were downloaded from the datasets provided by OnClass[73]. We used scripts in OnClass (v1.9.1) (https://www.nature.com/articles/s41467-021-25725-x) to perform the single dataset cross-validation, cross-dataset validation and calculate the AUROC values of classification.

We embedded textual descriptions of Cell Ontology terms using PubMedBERT. When embedding the gene expression features, the hidden dimension of fully connected layers was set to 30. After training, BioTranslator could automatically annotate a new cell to any new cell type. In each dataset, cells were split into the training and test set by setting different proportions of unseen cell types ranging from 10 to 90%. We compared our method to Support Vector Machine with rejection (SVM (reject)), Logistic Regression with rejection (Logistic Regression (reject)) and ACTINN. They were the best performing methods according to a previous benchmarking study[92]. SVM (reject) and Logistic Regression (reject) could classify a cell to the unknown type. We used a threshold of 0.7 to determine whether a cell was unknown. For a cell classified into unknown, we randomly set the probability of this cell belonging to one specific unseen cell type to 1.

Otherwise, the probability of that cell being classified to any unseen cell type was 0. ACTINN had a three layer neural network architecture, where the dimension of hidden layers were 100, 50, and 25. In the training process of our method and three baselines, we consistently set the learning rate to 0.0001, the batch size to 128, and the number of training epochs to 15.

BioTranslator could identify marker genes in two ways. In the first approach, BioTranslator calculates the Spearman correlation between expression values of one gene in different cells and the probability of these cells being classified into one cell type. The Spearman correlation was the probability of identifying this gene as the marker gene of the cell type. Since BioTranslator could annotate new cell types, it was capable of identifying marker genes for unseen cell types. Then we can calculate the AUROC of identifying marker genes for seen cell types and unseen cell types in each dataset. In the second approach, BioTranslator could identify marker genes without using the gene expression data. The capability of cross-modal translation enables BioTranslator to identify marker genes for cell types by mapping both the cells and genes into the same low-dimensional space. For a cell type, we calculated its low-dimensional representation by passing its textual description to the text encoder of BioTranslator. For a gene, we used the model trained on the Gene Ontology to obtain its low-dimension representation. Let the cell type embedding be $\mathbf{Y}_{CL}$ and the gene feature embedding be $\mathbf{F}_{gene}$. We calculated the probability of each candidate gene being a marker gene as:

$$p_{gene,CL} = \frac{1}{1 + e^{-(\mathbf{F}_{gene})^{\mathbf{T}}\mathbf{WY}_{CL}}}. \tag{5}$$

$\mathbf{W}$ was trained on the GOA datasets. We then used a binarized vector as the ground truth for each cell type. A dimension in this vector was 1 only when the corresponding gene was an identified marker. To evaluate the accuracy of marker gene identification for a cell type, we calculated the AUROC using the coefficients and the binarized vector. We also performed the cross-dataset evaluation on Tabula Microcebus, Tabula Sapiens, Tabula Muris (FACS), Tabula Muris (Droplet) and Lemur 1-4. We trained BioTranslator on one dataset to annotate cell types to cells in other datasets. Similarly to OnClass, we only considered genes that appear in both the training dataset and the test dataset in the cross-dataset validation. In the cell type marker gene network visualization, we find the enriched GO term for each community. We first obtained embeddings using BioTranslator for each node in the community and averaged over them to obtain a community embedding. Then we annotate the community using the GO term that has the largest similarity in terms of textual embeddings. We further restricted the number of annotated genes of this GO term to be ranged from 100 to 2500. If not possible, we will search for the nearest ancestor and descendant nodes on the GO hierarchy tree to find a valid GO term. The visualization was performed using Cytoscape (v3.9.1)[93].

## Unsupervised cross-modal prediction

Our multilingual translation framework enables predictions between different modalities without seeing any paired data. In our experiments, we investigated four modalities, including genes, drugs, phenotypes, and pathways. For each of these four modalities, we trained a separate encoder to translate it into the embedding space. Each translator is trained using pairs of biological features and textual descriptions. Each translator has an encoder. For these four translators, the encoders (i.e., from a biological modality to the embedding space) are different. In contrast, the embedding spaces are shared and guided by the text encoder, which is the PubMedBERT encoder fine-tined on various ontologies. Thus, our multilingual translation framework has five "languages": drug SMILES features, phenotype network-based features, gene network-based features, pathway gene-based features, and text.

We used protein network features, implemented using Mashup[49], as biological features of genes. We downloaded gene textual descriptions from the STRING database[88]. We only considered genes that appeared in both STRING and pretrained Mashup embeddings, and finally obtained 15,835 pairs of gene textual descriptions and gene Mashup embeddings. We used a two-layer MLP as the BioTranslator model architecture. After training the proposed BioTranslator model for 30 epochs, we obtained the BioTranslator gene-to-text translator that translates genes into text embedding space. The pathway-to-text translator was obtained by pooling the embeddings of genes within that pathway. We used sum pooling instead of mean pooling to consider the size of each pathway.

The drug-to-text translator was trained using drug SMILES features and drug textual descriptions. We obtained the paired SMILES features and textual descriptions from ChEBI-20[94] and excluded drugs that appeared in the test stage to avoid data leakage. We obtained a total number of 32,967 pairs. We used the Transformer encoder as the drug encoder, which is initialized using ChemBERTa[95,96]. We trained the BioTranslator model for 5 epochs with the start learning rate as 3e−5. We selected the linear schedule with 400 warmups for the learning rate.

The phenotype-to-text translator was trained using phenotype network features and phenotype textual descriptions. Each phenotype corresponds to one node in the Human Phenotype Ontology[97]. We constructed one network based on the Human Phenotype Ontology and obtained phenotype features based on the network topology. Specifically, we applied Mashup to the constructed graph and obtained network-based feature embeddings for each phenotype. We used the 'def' field in Phenotype Ontology as the phenotype textual description. We finally obtained 17,575 pairs of phenotype textual descriptions and network-based features. To avoid data leakage, we randomly chose 50% of the phenotypes as the training data, while the other 50% were used only at the test stage and their textual descriptions were never used.

We evaluated cross-modal predictions on three downstream tasks: predicting the targets of a new drug (Gene2Drug), predicting the genes of a new phenotype (Gene2Phenotype), and predicting the pathway of a new phenotype (Pathway2Phenotype). On the Gene2-Drug task, we used benchmarks from GDSC[46,98,99] and STITCH[45,100]. GDSC contains 132 drugs and STITCH contains 282,354 drugs. We collected the Gene2Phenotype data from Monarch[48,101] and obtained 4973 phenotypes that were never seen by the phenotype encoder. Pathways were obtained from Reactome and we obtained 1554 pathways. We didn't evaluate the pathway to drug association as there lacks a large-scale paired benchmark on it. Notably, we didn't use any paired data between two non-text modalities to train our multilingual system. For example, to predict drug target interaction, we used the gene to text encoder to translate genes into the embedding space and used the drug to text encoder to translate drugs to the same embedding space. We then predicted the probability of classifying a gene into the drug using the dot product between the two embeddings. We used the same approach to predict the tasks of Gene2Phenotype and Pathway2Phenotype. We used the NLTK python package[102] for word tokenization when finding overlapping words. To focus more on biomedical-related words, we removed English stop words using NLTK and also removed the 300 most common words across all pathway descriptions.

## Reporting summary

Further information on research design is available in the Nature Portfolio Reporting Summary linked to this article.

## Data availability

The datasets used for protein function prediction and pathway analysis are available at: https://figshare.com/articles/dataset/Protein_Pathway_data_tar/20120447. The processed datasets for cell type classification are available at: https://figshare.com/ndownloader/files/

28846647 and https://figshare.com/ndownloader/files/31777475. Other datasets used for single cell analysis are obtained from OnClass. The gene to text association file is obtained from https://stringdb-static.org/download/protein.info.v11.5/9606.protein.info.v11.5.txt.gz, the drug to text association files is available at: https://raw.githubusercontent.com/blender-nlp/MolT5/main/ChEBI-20_data/train.txt, the phenotype to text file is available at: https://raw.githubusercontent.com/obophenotype/human-phenotype-ontology/master/hp.obo. The GDSC dataset and STITCH dataset can be found at: https://www.cancerrxgene.org/downloads/drug_data and http://stitch.embl.de/cgi/download.pl?UserId=Moo0FJAHG6tW&sessionId=5ovagFIIUQjn. The gene to phenotype association file and pathway to phenotype association file are at: https://data.monarchinitiative.org/latest/tsv/all_associations/index.html. Source data are provided with this paper.

## Code availability
BioTranslator code is available at https://github.com/HanwenXuTHU/BioTranslatorProject and https://doi.org/10.5281/zenodo.7439647.

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

## Acknowledgements

R.B.A. is supported by NIH GM1023654 and Chan Zuckerberg Biohub.

## Author contributions

The authors confirm the contribution of this research as follows: H.X. and S.W. developed the conceptual ideas and designed the study; H.X. implemented the methods and analyzed the results; H.X., S.W., and A.W. wrote the manuscript; H.P. and R.B.A. provided the feedback.

## Competing interests

The authors declare no competing interests.
