## [Peer Review File · Nature Communications]

Multilingual Translation for Zero-shot Biomedical Classification using BioTranslatorREVIEWER COMMENTS

Reviewer #1 (Remarks to the Author):

This paper proposed BioTranslator, an extension of ProTranslator, for cross-model translation using the textual description to identify biological data instances. Experiments show that the tool outperforms ProTranslator, on new cell type discovery and pathway analysis.

In the beginning, I found the motivation of this paper interesting. However, the enthusiasm quickly faded when I discovered that the paper is a small extension of a different paper, "ProTranslator: zero-shot protein function prediction using textual description (published by RECOMB in May 2022)" from the same authors.

The key idea from the two papers - translating the text description to non-text biological data to enable zero-shot classification - are the same. The technological solutions are similar. At the same time, I appreciate that the authors demonstrated the generalizability of the model by fine-tuning the model on larger datasets and evaluating it in more experimental settings.

I really like the work of ProTranslator, but I feel the findings in this paper, compared to ProTranslator, are incremental and do not further advance the field, and thus do not meet the standards in this field and to be published in NC. I would suggest transferring this manuscript to Scieifiic Reports.

Reviewer #2 (Remarks to the Author):

This paper introduces an interesting work BioTranslator that performs cross-modal zero-shot learning on ontological terms and biological entities (genes, cells, pathways) to achieve the prediction of novel ontological terms and molecules. The authors conducted extensive experiments to show the generalization of the proposed method to other recent ones and the potential for biological discovery. Overall, the proposed method is interesting and showed its value for biomedical data analysis and discovery, especially for data from new classes, while most previous solutions mainly rely well defined labels. The main idea is borrowed from the heavily studied zero-shot learning that mapps different domain data toward an intermediate embedding space and align them to induce a zero-shot classifier. Although the experiments show the promising results to other typical solutions, I still have several comments for the authors:

1. The idea of embedding text and non-text data (image/video) toward an intermediate space for zero-shot learning is well studied, the performance improvement is mainly from PubmedBert? or from various ontologies?

2. How about the proposed method compared with other zero-shot learning solution with structure labels for zeros-shot learning for the similar problem?

[1]. Lee, C. W., Fang, W., Yeh, C. K., & Wang, Y. C. F. (2018). Multi-label zero-shot learning with structured knowledge graphs. In Proceedings of the IEEE conference on computer vision and pattern recognition (pp. 1576-1585).

[2]. Ou, G., Yu, G., Domeniconi, C., Lu, X., & Zhang, X. (2020). Multi-label zero-shot learning with graph convolutional networks. *Neural Networks*, 132, 333-341.

3. By the way, some protein function prediction solutions can make both few-shot and zero-shot learning based function prediction by referring to the structure of GO, even the annotations of a particular GO terms are not available, it still can associate related genes to this GO term.

[1]. Yu, G., Zhu, H., Domeniconi, C., & Liu, J. (2015). Predicting protein function via downward random walks on a gene ontology. *BMC bioinformatics*, 16:1.

4. How about the proposed method handling 5 or more new classes at the same time, now the results are mainly performed on leave one out fashion, which is too optimised to give the right solution.

5. How the proposed method borrows information from multiple ontologies or achieve knowledge transfer from text and biological entities can be further clarified. The figure 1 gives the main idea, how about the mathematical formulations?

Overall, the paper is interesting and can add values to the general biological data mining and discovery.

Response to Referees

BioTranslator: Multilingual Translation for Zero-shot Biomedical Classification

Hanwen Xu¹, Addie Woicik¹, Hoifung Poon², Russ B. Altman^{3,4,5}, Sheng Wang^{1#}

¹School of Computer Science and Engineering, University of Washington, Seattle, WA

²Microsoft Research, Redmond, WA

³Department of Bioengineering, Stanford University, Stanford, CA

⁴Department of Genetics, Stanford University, Stanford, CA

⁵Chan Zuckerberg Biohub, San Francisco, CA

[#]Email: swang@cs.washington.edu

We thank the referees for their valuable comments and evaluation of our work. In response to the referees' comments, we have conducted new experiments and analyses in our new manuscript. We have highlighted our revision in the new manuscript. Please see our point-to-point responses below.

Response to Referee #1:

• **Comments:** This paper proposed BioTranslator, an extension of ProTranslator, for cross-model translation using the textual description to identify biological data instances. Experiments show that the tool outperforms ProTranslator, on new cell type discovery and pathway analysis. In the beginning, I found the motivation of this paper interesting. However, the enthusiasm quickly faded when I discovered that the paper is a small extension of a different paper, "ProTranslator: zero-shot protein function prediction using textual description (published by RECOMB in May 2022)" from the same authors. The key idea from the two papers - translating the text description to non-text biological data to enable zero-shot classification - are the same. The technological solutions are similar. At the same time, I appreciate that the authors demonstrated the generalizability of the model by fine-tuning the model on larger datasets and evaluating it in more experimental settings. I really like the work of ProTranslator, but I feel the findings in this paper, compared to ProTranslator, are incremental and do not further advance the field, and thus do not meet the standards in this field and to be published in NC. I would suggest transferring this manuscript to Scientific Reports.

Answers: We thank the referee for the positive comments on our manuscript. In this revision, we have included a new result section, which conducts new analyses and experiments to clarify the differences between ProTranslator and BioTranslator. We highlight the key differences here:

- Both ProTranslator and BioTranslator can be regarded as a machine translation framework. ProTranslator is a **bilingual** translation framework, which can only map one kind of biological data (e.g., protein sequence) to text. In contrast, BioTranslator is a **multilingual** translation framework, which maps all kinds of biological data, such as gene expression, protein sequence and pathway graph to text.
- In traditional machine translation research, a multilingual translation system is a huge innovation over a bilingual translation system because 1) it can effectively utilize data from many languages, 2) it critically enables the translation between two languages that do not have any paired data, 3) it enables the translation of a low-resource language (e.g., an endangered language). In addition to the conceptual novelty, the multilingual translation system is also more challenging to develop, which further introduces technical and experimental contributions.

- Likewise, we believe that BioTranslator, as the first multilingual translation system, is unique and novel compared to ProTranslator. Specifically, BioTranslator connects all kinds of biological data modalities using text and enables the translation between two non-text modalities without paired data between them. This enables us to perform analyses that have never been done before and cannot be achieved by ProTranslator. For example, in our first submission, we showed how BioTranslator can identify cell type marker genes using gene description and cell type description, without using any cell type marker gene association pairs. This kind of analysis cannot be achieved by ProTranslator.
- To further highlight this difference, we have illustrated the power of a multilingual translation system (see **Section BioTranslator enables prediction between modalities without paired data**). This system includes five “languages”: drug SMILES-based features, gene network-based features, pathway network-based features, phenotype network-based features, and text. Among all the possible 10 language pairs, we trained this system using only 4 pairs: drug to text, gene to text, pathway to text, and phenotype to text. We then test on 3 pairs that have large-scale paired benchmark datasets: drug to gene, phenotype to gene, and phenotype to pathway. We observed very promising performance by BioTranslator, which even outperformed supervised approaches that used paired data on certain tasks. Our new results have thus shown that text data can effectively bridge all modalities, resulting in analyses that cannot be achieved by ProTranslator.

In addition to the new analyses in the result section, we have added a paragraph in the discussion section to clarify the difference.

Response to Referee #2:

• **Comments:** This paper introduces an interesting work BioTranslator that performs cross-modal zero-shot learning on ontological terms and biological entities (genes, cells, pathways) to achieve the prediction of novel ontological terms and molecules. The authors conducted extensive experiments to show the generalization of the proposed method to other recent ones and the potential for biological discovery. Overall, the proposed method is interesting and showed its value for biomedical data analysis and discovery, especially for data from new classes, while most previous solutions mainly rely well defined labels. The main idea is borrowed from the heavily studied zero-shot learning that maps different domain data toward an intermediate embedding space and align them to induce a zero-shot classifier. Although the experiments show the promising results to other typical solutions, I still have several comments for the authors.

Summary of revision: We thank the Referee for the positive comments on BioTranslator. In this revision, we have conducted new experiments and analyses as well as comparing BioTranslator to existing approaches suggested by the Referee. The substantial improvement of BioTranslator again confirmed our belief that BioTranslator can serve as an effective approach for zero-shot classification in biomedicine.

• **Question 1:** The idea of embedding text and non-text data (image/video) toward an intermediate space for zero-shot learning is well studied, the performance improvement is mainly from PubmedBert? or from various ontologies?

Answer: Compared to existing works that co-embed text and image or video data, BioTranslator tackles a novel problem of co-embedding text data and biological data, such as gene expression, pathways and protein sequences. We found that the improvement comes from both using PubMedBERT and fine-tuning the model on 225 ontologies using contrastive learning. The comparison approach ProTranslator also utilized PubMedBERT, but didn't fine-tune on various ontologies. Our improvement over ProTranslator verifies the importance of using various ontologies (**Fig. 2**). The comparison approach TF-IDF and Word2Vec didn't use PubMedBERT. Our improvement over them demonstrates the importance of using PubMedBERT (**Fig. 2**).

• **Question 2:** How about the proposed method compared with other zero-shot learning solution with structure labels for zeros-shot learning for the similar problem?

[1]. Lee, C. W., Fang, W., Yeh, C. K., & Wang, Y. C. F. (2018). Multi-label zero-shot learning with structured knowledge graphs. In Proceedings of the IEEE conference on computer vision and pattern recognition (pp. 1576-1585).

[2]. Ou, G., Yu, G., Domeniconi, C., Lu, X., & Zhang, X. (2020). Multi-label zero-shot learning with graph convolutional networks. *Neural Networks*, 132, 333-341.

Answer: We have now cited and included those two comparison approaches in our new version (**Fig. 2 a-e, Supplementary Fig. 2-6**). We found that ML-ZSL outperformed other graph-based approaches, indicating the effectiveness of information propagation mechanism learning process. We found that BioTranslator outperformed both approaches. We attributed this to BioTranslator's ability to consider various ontologies from diverse domains. We have now added our new results and analyses (**Fig. 2 a-e, Supplementary Fig. 2-6**).

• **Question 3:** By the way, some protein function prediction solutions can make both few-shot and zero-shot learning based function prediction by referring to the structure of GO, even the annotations of a particular GO terms are not available, it still can associate related genes to this GO term.

[1]. Yu, G., Zhu, H., Domeniconi, C., & Liu, J. (2015). Predicting protein function via downward random walks on a gene ontology. *BMC bioinformatics*, 16:1.

Answer: Thank you for referring this paper to us. We have now cited this paper. We have compared our approach to clusDCA, which exploits the ontology graph to enable predictions on unseen classes. After investigating dRW, we found that it was not applicable to our setting. dRW is designed to predict the missing functions of a protein given when it has been annotated by several GO terms, while our setting is to predict functions for a totally new protein, without any existing annotations to start random walks. We also now cited and discussed dRW in our new manuscript.

• **Question 4:** How about the proposed method handling 5 or more new classes at the same time, now the results are mainly performed on leave one out fashion, which is too optimised to give the right solution.

Answer: We apologize for the confusion. In both our protein function prediction task and single cell analysis, we have tested on more than 5 new classes for each validation set. For example, in the protein function prediction cross validation on GOA (Human), we evaluate on more than 4200 unseen classes at once for each fold. Likewise, in the cell type discovery task on Tabula Microcebus, more than 61 new cell types are evaluated at the same time when the unseen ratio is 0.5. We have now further clarified this in the manuscript to avoid future confusion.

• **Question 5:** How the proposed method borrows information from multiple ontologies or achieve knowledge transfer from text and biological entities can be further clarified. The figure 1 gives the main idea, how about the mathematical formulations?

Answer: We have now added mathematical formulations in our new manuscript in the Method Section. Our goal is to estimate the probability of classifying one instance with feature F into the unseen class with textual description embedding Y_{unseen} , that is, to approximate $P(Y_{unseen}|F)$. We can expand it as:

$$P(Y_{unseen}|F) = P(Y_{unseen}|Y_{seen})P(Y_{seen}|F),$$

where fitting $P(Y_{seen}|F)$ denotes the learning process of our bi-nonlinear model. Then fine-tuning the text encoder on multiple ontologies corresponds to estimate $P(Y_{unseen}|Y_{seen})$, where classes with similar textual descriptions tend to be co-located in the embedding space.

REVIEWERS' COMMENTS

Reviewer #2 (Remarks to the Author):

The authors mostly address my technical concerns and more clearly explain the usage and application value of BioTranslator.